# Analyzing Facial Asymmetry in Alzheimer’s Dementia Using Image-Based Technology

**DOI:** 10.3390/biomedicines11102802

**Published:** 2023-10-16

**Authors:** Ching-Fang Chien, Jia-Li Sung, Chung-Pang Wang, Chen-Wen Yen, Yuan-Han Yang

**Affiliations:** 1Department of Neurology, Kaohsiung Municipal Ta-Tung Hospital, Kaohsiung Medical University Hospital, Kaohsiung Medical University, Kaohsiung 80756, Taiwan; 2Department of Neurology, Kaohsiung Medical University Hospital, Kaohsiung Medical University, Kaohsiung 80756, Taiwan; 3Department of Mechanical and Electromechanical Engineering, National Sun Yat-sen University, Kaohsiung 80424, Taiwan; 4Department of and Master’s Program in Neurology, Faculty of Medicine, College of Medicine, Kaohsiung Medical University, Kaohsiung 80708, Taiwan; 5Neuroscience Research Center, Kaohsiung Medical University, Kaohsiung 80708, Taiwan

**Keywords:** Alzheimer’s dementia, facial asymmetry, facial landmarks, accelerated aging, Procrustes method

## Abstract

Several studies have demonstrated accelerated brain aging in Alzheimer’s dementia (AD). Previous studies have also reported that facial asymmetry increases with age. Because obtaining facial images is much easier than obtaining brain images, the aim of this work was to investigate whether AD exhibits accelerated aging patterns in facial asymmetry. We developed new facial asymmetry measures to compare Alzheimer’s patients with healthy controls. A three-dimensional camera was used to capture facial images, and 68 facial landmarks were identified using an open-source machine-learning algorithm called OpenFace. A standard image registration method was used to align the three-dimensional original and mirrored facial images. This study used the registration error, representing landmark superimposition asymmetry distances, to examine 29 pairs of landmarks to characterize facial asymmetry. After comparing the facial images of 150 patients with AD with those of 150 age- and sex-matched non-demented controls, we found that the asymmetry of 20 landmarks was significantly different in AD than in the controls (*p* < 0.05). The AD-linked asymmetry was concentrated in the face edge, eyebrows, eyes, nostrils, and mouth. Facial asymmetry evaluation may thus serve as a tool for the detection of AD.

## 1. Introduction

Alzheimer’s dementia (AD) is characterized by a gradual decline in cognitive functions, particularly memory, impacting spatial awareness, judgment, and causing neuropsychiatric symptoms, and thus significantly affecting daily life. The prevailing understanding of AD pathology involves the accumulation of β-amyloid plaques outside nerve cells and tau protein tangles inside nerve cells, leading to neuronal damage and death [1]. However, these pathological changes are considered late-stage manifestations, and there are few hypotheses regarding the events preceding AD onset. In AD, synaptic degeneration can be detected in the early stages of pathological progression before neuronal degeneration. Synapses are where β-amyloid peptides are generated and serve as targets for toxic β-amyloid oligomers [2]. Neuroinflammation, which has been confirmed to play a pivotal role in AD, involves various inflammatory processes within the central nervous system [3]. Activated immune cells release inflammatory molecules in the brain, potentially causing neuronal damage. This inflammation may be a consequence of neuronal damage and abnormal protein metabolism, contributing to the progression of the disease. Among innate immune cells, microglial cells are central participants in neuroinflammation. Activated microglial cells exhibit diverse cellular profiles and interact with β-amyloid and tau proteins, as well as neuronal circuits in various ways. In later AD stages, excessive neuroinflammation leads to neuronal and glial cell death. Aβ clearance is impaired, and microglial cells release high levels of pro-inflammatory molecules, causing neurodegeneration. This creates a vicious cycle, and the release of ROS and NOS by microglial cells accelerates neuronal loss [3,4]. Vascular dysregulation may also be associated with AD. In post-mortem studies, 60–90% of clinical diagnosed AD patients exhibit brain vascular abnormalities, with 30% showing signs of cerebral infarctions [5]. About one-third of clinical diagnosed vascular dementia patients present AD-related pathological features, including amyloid plaques and neurofibrillary tangles [6]. Genetic factors play a role, particularly in early-onset AD with mutations in genes like APP, PSEN1, and PSEN2. However, most AD cases are late-onset and involve complex interactions between multiple genetic and environmental factors. From a macroscopic perspective, AD primarily affects brain regions such as the cerebral cortex and the hippocampus, resulting in atrophy and volume loss. Owing to the limited treatability of AD currently, earlier detection of AD and earlier use of medications such as acetylcholinesterase inhibitors is important as this may postpone disease progression and preserve patients’ activities of daily living.

The developmental processes of the brain and facial morphology are highly interconnected [7,8]. During early embryonic development, the neuroepithelial cells located within the neural folds gives rise to facial precursor cells known as Cranial Neural Crest Cells (CNCC) [9]. These CNCCs undergo a transformation from an epithelial to a mesenchymal state and migrate ventrally, playing a significant role in the development of a majority of craniofacial bones and connective tissues. Early brain growth impacts facial prominence positioning and growth. In proximity to the CNCCs is the forebrain, which encompasses the cerebral cortex. The CNCC-secreted paracrine factors also regulate brain development. These interactions are well-documented in animal embryo studies and align with the co-occurrence of neurodevelopmental and craniofacial abnormalities in humans [8].

Many studies have demonstrated a pattern of accelerated brain aging in patients with AD. Based on MRI imaging, the BrainAGE algorithm linked AD to accelerated brain aging compared with a cognitively normal group and a group with mild cognitive impairment (MCI) [10,11,12,13]. Huang et al. utilized MRI and a machine learning model to predict brain age based on gray matter volume, revealing an accelerated brain aging trajectory in patients with MCI compared with healthy individuals [14]. Linden et al. found that facial asymmetry increases with age [15], and normal face development is highly contingent on normal brain development [8,16,17]. Facial malformation was also observed to be related to underlying brain disease [18]. Face–brain asymmetry has been identified in autism spectrum disorders [19] and in patients with focal epilepsy [20]. The etiology of facial asymmetry is multifactorial, including functional, neuromuscular, stomatognathic, environmental, congenital, traumatic, and neoplastic factors, as well as the effects of chronic skin diseases and aging. The face reveals clues to sex, age, and genetics. One study reported that AD patients have an “older-appearing face” based on the estimation of an independent panel of eight human raters [21]. Penke et al. also found that facial asymmetry was an important predictor of cognitive decline in older persons [22].

Most diagnostic procedures for AD are invasive or time consuming. For example, psychological assessment is time consuming, cerebrospinal fluid examination is invasive, and amyloid positron emission tomography is costly [23]. Beyond the conventional chemical and imaging biomarkers, other markers for AD detection are being developed [24,25]. As far as we know, there are no facial presentations that are typical of AD.

Review articles have appeared that cover many single-face image-based methods for estimating age [26,27,28]. However, these articles do not focus on the relationship between face aging and neurodegenerative diseases such as AD. In an initial investigation of the impact of AD on the human face, the above-cited literature points to facial symmetry as an interesting feature to focus on.

Considering that obtaining facial images is much easier than obtaining brain images, the aim of this study was to investigate whether facial asymmetry is potentially a marker for differentiating patients with AD from age- and sex-matched, healthy control subjects.

## 2. Materials and Methods

### 2.1. Participants

A total of 300 Taiwanese participants, including 150 AD patients and 150 controls without dementia, were recruited at Kaohsiung Municipal Ta-Tung Hospital in southern Taiwan. Participants had no previous facial trauma or disease. The diagnosis of AD was based on the NINCDS-ADRDA criteria [29] with reference to a series of comprehensive neuropsychological tests, including the Mini-Mental State Examination (MMSE) [30] derived from the Cognitive Abilities Screening Instrument (CASI) [31], Neuropsychiatric Inventory (NPI) [32], and Clinical Dementia Rating (CDR) scale [33]. The neuroimaging and blood check to exclude other conditions possibly contributing to the diagnosis of dementia were conducted simultaneously. The control group without dementia was recruited by a senior neurologist to exclude dementia and other systemic diseases.

This study was approved by the institutional review board of the Kaohsiung Medical University Chung-Ho Memorial Hospital (KMUHIRB-SV(I)-20210067). All patients gave written consent for the use of their images for research purposes before the image was taken.

### 2.2. Data Collection

#### 2.2.1. Image Retraction

The images of patients were recorded with a stereo camera (Intel RealSense D435) to capture three-dimensional (3D) facial images. With a 30 Hz frame rate, every participant took thirty images. The resolution of the infrared camera is 1280 × 720. The resolution of the RGB camera is 1920 × 1080. Under standardized lighting conditions and against a constant background, test subjects were guided by a clinician to a natural head position at a distance of about 40 to 60 cm from the 3D camera. While keeping the Frankfort horizontal (FH) plane parallel to the floor subjects were asked to maintain a neutral facial expression with both eyes looking forward. The experiments were carried out in a quiet room of Kaohsiung Municipal Ta-Tung Hospital and subjects were required to stay seated during the test. Subjects were instructed to show a natural and relaxed facial expression while the images were taken. After removing images with closed-eyes or other inappropriate expressions, we included those with adequate quality for further analysis.

#### 2.2.2. Face Landmark Detection and Pre-Processing

For each of the 3D face images analyzed in this study, we used OpenFace [34], a facial behavior analysis toolkit, to detect 68 facial landmarks. OpenFace’s open-source availability, robust facial landmark detection, and strong performance make it a preferred choice for research and development in the field of facial analysis. As shown in Figure 1 and Figure 2, as a machine learning algorithm, OpenFace detects 29 hemiface landmarks for each side of the face and 10 midface landmarks. Each landmark is represented by its 3-dimension *x* (horizontal), *y* (vertical), and *z* (depth) coordinates. Since OpenFace also provides the yaw angle for the human face, we took the face with the smallest yaw angle as the frame for subsequent calculation (Figure 3).

In measuring the position of the facial landmarks, inaccurate results were frequently encountered in estimating the depth of facial contour landmarks since these landmarks are located in the boundary between face and background regions. To remedy this problem, by processing the depth measurements provided by the camera system, we used K-means clustering algorithm [35] to automatically classify image pixels into a foreground (face) cluster and a background cluster. For each of the facial landmarks that were incorrectly assigned into the background cluster, we used the following rules to replace its depth measurement. First, with such a landmark as the center, we set up a 7 × 7 pixel window. Next, we examined how many pixels within this window were clustered into the facial cluster and used the average depth of these pixels as the depth of the landmark. When pixels of this 7 × 7 window were all assigned into the background cluster, the landmark depth was chosen as the median value of the depth of all 68 landmarks.

The size of a face image varies with the actual face size and the distance between the subject and the camera. To develop asymmetry measures that are invariant to these variations, we first calculated the centroid of all facial landmarks and denoted its coordinates as ($\bar{x}$, $\bar{y}$). Next, we used Equation (1a) to determine the horizontal normalized coefficient *d_x_* which is the average of all the horizontal distances between the landmark centroid and landmarks. Similarly, this work used Equation (1b) to determine the vertical normalized coefficient *d_y_*. Note that in the *xy* plane, distances are measured in pixels whereas distances in the *z* direction are measured in mm. This unit inconsistency problem will be addressed in the following section.
(1a)$$\sum_{i=1}^{68}\frac{\left|x_{i}-\bar{x}\right|}{68}=d_{x}\;$$
(1b)$$\sum_{i=1}^{68}\frac{\left|y_{i}-\bar{y}\right|}{68}=d_{y}$$

Note that the above normalization procedure was not employed in the depth direction. There are two reasons for such an arrangement. The first is that, in the depth direction, the differences between the landmarks are invariant to the distance between the subject and the camera. The second reason is that, as shown in the next section, the proposed asymmetry measure does not account for depth direction differences.

### 2.3. Normalization and Procrustes Analysis of the Facial Coordinates

This work characterizes facial asymmetry by comparing the location of each of the 29 right hemiface landmarks to the location of the corresponding left hemiface landmarks. To achieve this goal, the basic idea of the proposed approach is first to create a mirror face image by horizontally flipping the original image and then uses the Procrustes method [36,37] to align the mirror and original images. The Procrustes method, used in shape analysis and image alignment, aligns objects like facial images by translation, rotation, scaling, and reflection to minimize differences. It places facial landmarks in a shared coordinate system and orientation. This is employed to create a mirrored right-face image for LSAD calculations with the left face. Note that since the right hemiface of the mirror image is essentially the left hemiface of the original image, the Procrustes method only needs to align the right hemiface of the original and mirror images. This alignment procedure is critical for accurate asymmetry assessment since it is very difficult to perfectly orient the test subjects toward the camera.

To implement the proposed approach, we first denoted the coordinates of the *i*th right hemiface landmarks of the original image and mirror image as (*x_i_*, *y_i_*, *z_i_*) and (*α_i_*, *β_i_*, *γ_i_*), respectively. The coordinates of the centroids of the right hemiface landmarks of the original and mirror images are denoted as ($\bar{x}$, $\bar{y},\;\bar{z}$) and ($\bar{\alpha }$, $\bar{\beta },\;\bar{\gamma }$), respectively. Based on the normalization concept introduced in the previous section, we used Equation (2a) to calculate the normalized horizontal coordinate of the *i*th right hemiface landmarks of the original image. In an identical manner, we used Equation (2b) to determine the normalized vertical coordinate of the *i*th right hemiface landmarks of the original image. Similarly, for the mirror image, Equation (3a,b) were used to determine the normalized horizontal and vertical coordinates of *i*th right hemiface landmarks of the mirror image. For the depth direction, this work used Equations (2c) and (3c) to determine the normalized depth coordinates of the right hemiface landmarks of the original image and mirror image, respectively. Note that the depth coefficient *c* of Equations (2c) and (3c) was introduced to adjust the relative weighting between the alignment errors of the *xy* plane and the depth direction in applying the Procrustes method and at the same time resolving the unit inconsistency problem of the *xy* plane and the *z* direction.
(2a)$$X_{i}^{}=\left(x_{i}^{}-{\bar{x}}^{}\right)/d_{x}^{}$$
(2b)$$Y_{i}^{}=\left(y_{i}^{}-{\bar{y}}^{}\right)/d_{y}^{}$$
(2c)$$Z_{i}^{}=c*\left(z_{i}^{}-{\bar{z}}^{}\right)$$
(3a)$$A_{i}^{}=\left(\alpha_{i}^{}-{\bar{\alpha }}^{}\right)/d_{x}^{}$$
(3b)$$B_{i}^{}=\left(\beta_{i}^{}-{\bar{\beta }}^{}\right)/d_{y}^{}$$
(3c)$$C_{i}^{}=c*\left(\gamma_{i}^{}-{\bar{\gamma }}^{}\right)$$

Once the depth coefficient *c* was determined, with the alignment error of the *i*th right hemiface quantified as (*X_i_* − *A_i_*)^2^ + (*Y_i_* − *B_i_*)^2^ + (*Z_i_* − *C_i_*)^2^, we used the Procrustes method to translate and rotate the original image so that the sum of squares of the alignment error of all right hemiface landmarks could be minimized. With ${X^{\prime }}_{i}^{}$ and ${Y^{\prime }}_{i}^{}$ denoting the horizontal and vertical normalized coordinate of the *i*th landmark of the aligned original image, we used Equation (4) to quantify the asymmetry associated with the *i*th landmark. Hereafter, these asymmetry measures will be referred to as landmark superimposition asymmetry distances (LSADs). The sum of LSAD is also proposed as an asymmetry measure (Equation (5)).
(4)$$LSAD_{i}^{}=\sqrt{{\left(A_{i}^{}-{X^{\prime }}_{i}^{}\right)}^{2}+{\left(B_{i}^{}-{Y^{\prime }}_{i}^{}\right)}^{2}}$$
(5)$$LSAD_{s}^{}=\sum_{i=1}^{29}LSAD_{i}^{}$$

With the asymmetry measures specified, the depth coefficient *c* can be determined by the following steps. First, we extensively tested different values of the depth coefficient *c*. For each of these tested *c* values, we calculated all the asymmetry measures of Equations (4) and (5) for all the test subjects. With these 30 asymmetry measures logged for all test individuals, we performed an independent sample one-tailed *t*-test between AD and controls and counted the number of asymmetry measures with significant differences. Because the number of asymmetry measures with significant differences is the most when the depth coefficient was 0.016, the value of the depth coefficient *c* was chosen as 0.016 for subsequent analysis.

### 2.4. Statistic Analysis

Continuous variables are expressed as the mean with the standard deviation, whereas categorical variables are presented as percentages. The association between the AD and controls and demographic and clinical characteristics (age, sex, CDR, CDR-SB, MMSE, and CASI) was explored using *t*-tests and the chi-square test. *p*-values for comparisons across groups of clinical and demographic characteristics were derived from the aforementioned analyses. The independent *t*-test is used to determine if the mean of the LSAD of the AD patients and controls is significantly different. All reported *p*-values are two-sided, with a *p*-value < 0.05 considered to be statistically significant. Analyses were performed using SPSS Version 26.

## 3. Results

### 3.1. Demographic Characteristics of Recruited Participants

Patients’ age ranged from 56 to 79 years (mean: 72.2 years; standard deviation: 4.9 years). Fifty-seven patients were male and ninety-three were female. The CDR sum of boxes score was 4.8 ± 2.9 (mean ± SD), the MMSE total score was 21.0 ± 4.4 (mean ± SD), and the CASI total score was 65.7 ± 14.2 (mean ± SD).

Healthy subjects were chosen as the control group (age: 52–93 years, mean: 71.8 years, standard deviation: 7.4 years) and included forty-six males and one hundred and four females. The CDR sum of boxes score was 1.5 ± 1.4 (mean ± SD), the MMSE total score was 24.5 ± 3.7 (mean ± SD), and the CASI total score was 83.7 ± 10.5 (mean ± SD). A summary of subjects’ characteristics is reported in Table 1.

### 3.2. The Comparison of Landmark Superimposition Asymmetry Distances (LSADs) of AD and Controls

Figure 4 shows the 68 landmarks, including 29 hemiface landmarks for each side of the faces and 10 midface landmarks. The landmark superimposition asymmetry distances (LSADs) of AD and controls are shown in Table 2, expressed as the mean with the standard deviation. Among the 29 LSADs of the AD patients, 20 of these LSADs are significantly larger than those of the controls. In face edge pair 1, the mean distance was 0.71 ± 0.51 for AD patients and 0.63 ± 0.53 for controls (*p* = 0.003). In face edge pair 2, the mean distance was 0.43 ± 0.30 for AD patients and 0.40 ± 0.37 for controls (*p* = 0.006). In face edge pair 3, the mean distance was 0.47 ± 0.34 for AD patients and 0.39 ± 0.33 for controls (*p* = 0.009). In face edge pair 4, the mean distance was 0.56 ± 0.37 for AD patients and 0.46 ± 0.37 for controls (*p* = 0.012). In face edge pair 5, the mean distance was 0.57 ± 0.39 for AD patients and 0.48 ± 0.37 for controls (*p* = 0.010). In face edge pair 6, the mean distance was 0.51 ± 0.33 for AD patients and 0.44 ± 0.32 for controls (*p* = 0.003). In face edge pair 7, the mean distance was 0.41 ± 0.31 for AD patients and 0.37 ± 0.31 for controls (*p* = 0.041). In face edge pair 8, the mean distance was 0.34 ± 0.35 for AD patients and 0.25 ± 0.24 for controls (*p* = 0.007). In eyebrows pair 9, the mean distance was 0.25 ± 0.19 for AD patients and 0.22 ± 0.15 for controls (*p* = 0.005). In eyebrows pair 10, the mean distance was 0.17 ± 0.12 for AD patients and 0.16 ± 0.12 for controls (*p* = 0.004). In eyebrows pair 13, the mean distance was 0.29 ± 0.20 for AD patients and 0.25 ± 0.18 for controls (*p* = 0.001). In nostrils pair 14, the mean distance was 0.37 ± 0.23 for AD patients and 0.31 ± 0.22 for controls (*p* = 0.013). In nostrils pair 15, the mean distance was 0.44 ± 0.25 for AD patients and 0.39 ± 0.26 for controls (*p* = 0.016). In eyes pair 16, the mean distance was 0.12 ± 0.10 for AD patients and 0.09 ± 0.06 for controls (*p* = 0.024). In eyes pair 17, the mean distance was 0.15 ± 0.12 for AD patients and 0.11 ± 0.09 for controls (*p* = 0.001). In eyes pair 18, the mean distance was 0.13 ± 0.10 for AD patients and 0.11 ± 0.09 for controls (*p* = 0.0004). In eyes pair 19, the mean distance was 0.12 ± 0.07 for AD patients and 0.10 ± 0.07 for controls (*p* = 0.0009). In mouth pair 22, the mean distance was 0.13 ± 0.09 for AD patients and 0.11 ± 0.08 for controls (*p* = 0.023). In mouth pair 24, the mean distance was 0.43 ± 0.23 for AD patients and 0.37 ± 0.23 for controls (*p* = 0.035). In mouth pair 26, the mean distance was 0.25 ± 0.16 for AD patients and 0.21 ± 0.15 for controls (*p* = 0.042). The sum of LSAD was 10.33 ± 4.39 for AD patients and 9.14 ± 4.45 for controls (*p* = 0.003). Other LSADs of the AD patients were also larger than those of the controls but did not show statistical significance.

## 4. Discussion

### 4.1. Key Findings

In this work, we aimed to develop a facial asymmetry assessment system to differentiate Alzheimer’s patients from non-dementia people. Our study showed, among the 29 tested landmark pairs, 20 pairs of LSADs of the AD patients are significantly larger than those of the controls, including all face edge landmarks, 3 eyebrow landmarks, all landmarks of the nostrils, 4 landmarks of the eyes, and 3 mouth landmarks.

### 4.2. Facial Differences in Ethnic Skin

Taiwan is located in East Asia and its population is primarily composed of Han Chinese and indigenous peoples. Also, there are subtle differences in the facial features of northeast Asian and southeast Asian ethnic groups, often reflecting mixed phenotypic characteristics influenced by their proximity to neighboring populations. Farkas et al. found that the most significant differences among different racial groups were observed in the eye socket area, nose height, and nose width. Middle Eastern and Asian populations had narrower intercanthal distances with less eye corner width, whereas Caucasians exhibited a narrower nasal base with a more pronounced nasal tip [38,39]. Facial aging processes are similar across races, but differences in skeletal support and soft tissue tendencies result in slower facial aging in Asians compared to Caucasians. Asians often have dense fat and fiber connections in the mid-face, reducing sagging and resulting in fewer superficial wrinkles [40]. Although our assessment method is not influenced by skin color, wrinkles, or nasal bridge height, it is important to acknowledge that racial differences have a significant impact on morphological variations related to facial asymmetry [41]. This could limit the comparability of facial asymmetry measurements across different ethnic groups, representing a limitation in our study.

### 4.3. Possible Biological and Neural Mechanisms of Facial Asymmetry in AD

Facial development begins in the early fourth week of embryonic development when the frontonasal process emerges. By the fifth week, the nasal processes and maxillary/mandibular processes appear. They grow and fuse, forming the foundation of facial development. This process is typically completed by the eighth week, defining three primary facial regions: the frontal, maxillary, and mandibular parts [42,43].

In human facial development, genetic factors and growth factors play crucial roles. Key pathways include Fibroblast Growth Factor (FGF), Hedgehog (HH), and Bone Morphogenetic Protein (BMP). FGFs are involved in precise regulation, with mutations linked to syndromes like Crouzon and Apert. The HH pathway influences neural crest cells and jaw development, and mutations in the Sonic Hedgehog (SHH) gene can cause midline defects and eye issues. BMP signaling is essential for craniofacial bone development and disruptions can lead to various anomalies, including a cleft palate and facial defects [7,8,16]. However, there is currently no direct evidence showing that these signaling molecules are physiologically relevant to the pathogenesis of AD.

AD causes cortical atrophy and brain function decline. Abbate et al. proposed a pathophysiological hypothesis for AD, suggesting shared spatial information in cortical arealization during development and AD [44,45]. Some AD subtypes affect specific brain regions, potentially leading to lateralization. However, limited literature explores this. Compared to other vertebrates, AD processes, like amyloid deposition due to metabolism and tau pathogenesis from adult neurogenesis/migration, are emphasized in the complex human cortex. The human longevity revolution likely contributes to these extremes.

The two most substantiated risk factors for AD are genetics and aging. Mutations in genes like PSEN1, PSEN2, and APP lead to early-onset familial AD, while late-onset sporadic AD results from a combination of factors, including aging, lifestyle, environment, and genetics [46]. APOE4 is the most significant genetic risk factor for late-onset AD, associated with earlier accumulation of amyloid plaques and neurofibrillary tangles. APOE gene variants directly impact brain development [47]. In a study with 1187 healthy children, APOE4 carriers had a thinner temporal cortex, smaller hippocampus, and weaker executive function correlations [48].

In normal aging, neuron loss is minimal, but there are changes in dendrite characteristics. These alterations vary across brain regions, with the most significant volume reduction occurring in specific areas like the frontal and temporal cortex [49]. Facial aging involves changes in bone structure, soft tissues, and skin, with interrelated effects. Bone resorption occurs with aging, affecting support, leading to soft tissue contraction and shifting. External factors like sun exposure and smoking impact skin aging, while collagen degradation accelerates and synthesis decreases with age, resulting in dermal atrophy [50,51,52].

Similar to our study, the BrainAGE algorithm via MRI has shown accelerated brain aging patterns in AD patients [10]. Dysfunctional DNA repair is also implicated in AD risk. While DNA methylation patterns generally decrease with age, specific gene regions may experience excessive methylation, resulting in increased variability. Epigenetic clocks, based on DNA methylation at specific sites, serve as reliable aging markers [53,54]. The difference between actual age and biological age is referred to as accelerated or decelerated epigenetic aging. Studies on dementia and mild cognitive impairment have yielded inconclusive results, with some suggesting a link between epigenetic age and dementia risk while others find no significant association [53,54].

Unlike our study, Naqvi et al. provided genetic and MRI evidence supporting the link between human facial and brain shape. They explored changes in various facial quadrants using GWAS but found no significant associations in AD [55]. This lack of association may be due to the complex etiology of AD, which involves a combination of genes and factors related to late-life plaque accumulation and neurodegeneration.

### 4.4. Method Comparisons

Recent studies used deep learning for age estimation from facial images [17,18,19] and showed promise. However, the connection between facial images and AD detection remains intriguing due to natural facial asymmetry variation. Quantifying facial asymmetry lacks a universally accepted standard [38]. This study introduces a landmark-based measure to address this issue.

Previous research has employed various methods to assess facial asymmetry, such as using the asymmetry distance calculated by root-mean-squared-error (RMSE) [56,57]. For instance, Ferrario et al. [58] utilized a camera-automatic three-dimensional landmark detection system, employing 16 landmarks, including 6 median points and 10 points representing the eyes, nose, mouth, and face edges. They applied Euclidean distance matrix analysis to compare the left and right hemifaces. Xiong et al. [59] analyzed 21 automatic facial landmarks, encompassing 7 medial points and 7 pairs of landmarks for the eyes, nose, and mouth. Ekrami et al. [60] examined 19 manual facial landmarks, consisting of 7 medial points and 6 pairs of landmarks for the eyes, nose, and mouth. In addition to biological landmarks, some studies incorporated nevi and wrinkles to enhance sensitivity. Many studies attempted to establish a reference frame for quantifying asymmetry, but determining the accurate midline frames posed another challenge [36,37,56]. To reduce errors, some studies employed multiple validation methods.

Regarding the assessment of facial asymmetry for facial palsy, there are various methods available. Gaber et al. employed Kinect V2 for automated recognition of facial features and created their own database. To address the issue of small sample sizes, they utilized undersampling and data augmentation for compensation [61]. Abayomi-Alli et al. introduced a deep learning-based method for facial paralysis detection and classification. They employed a novel image enhancement technique that extended random erasure enhancement using Voronoi subdivision structures. Subsequently, they used the SqueezeNet deep neural network for extracting deep features and employed a multi-class classifier for classification [62]. Wei et al. proposed a set of new facial features and augmented reality tools to assist users in interactively assessing facial symmetry. The development process involved four different datasets and focused on extracting features suitable for application on a smartphone program with lower computational costs [63].

After reviewing the previous literature, this study utilized a camera-automatic three-dimensional landmark detection system with OpenFace and analyzed facial asymmetry using the Procrustes method, which appeared to be the most effective and intuitive approach. The LSADs in our study indicated increased facial asymmetry in AD patients, particularly in regions such as the face edge, eyebrows, eyes, nostrils, and mouth. This method has been applied in other studies, albeit with different landmarks. Concerning age, Linden et al. demonstrated increased facial asymmetry in the lower two-thirds compared to the upper one-third in older age groups, while in younger age groups, more asymmetries were found in the upper third of the face [15]. Recently, Umeda et al. employed deep learning models to distinguish between the faces of dementia patients and non-dementia patients, with the lower face epochs providing better sensitivity and specificity. However, their results may have been influenced by the significantly older age of their AD group [23]. Greater normal variability in the global population was observed in the location of facial landmarks, including tragion, gonion, and zygion [64]. Zygion is close to our face edge landmark pair 2, and gonion is near our face edge landmark pair 5. Vertical eye movement differences have been found in AD patients compared to normal individuals [65], and abnormal oculomotor movement has been noted in several neurodegenerative diseases [66]. However, direct associations between eye shape or location and AD have not been identified in previous studies, nor have such associations been found for the rest of the facial landmarks.

### 4.5. Strengths

Facial asymmetry assessment is our initial step in evaluating facial features. While some conditions like stroke exhibit obvious facial asymmetry, its impact in chronic degenerative diseases is less discernible and is compounded by age-related effects. Thus, we employed a state-of-the-art machine learning algorithm to automatically and effectively identify facial landmarks, reducing human labor and enhancing the reproducibility of facial asymmetry measurements.

Compared to previous research, our study possesses several strengths. Firstly, by utilizing a 3D camera for facial capture, our study introduced a systematic approach to calibrate the relative orientation between the human head and the camera, leading to more accurate facial asymmetry computation. Secondly, to our knowledge, this is the first study to distinguish AD patients from non-dementia individuals through the objective evaluation of facial asymmetry. Thirdly, our study incorporated more reference points than other studies, allowing for the assessment of more subtle and precise features. Lastly, the included AD patients were primarily in the mild severity range (CDR: 0.5–1.0), indicating that our results may be applicable for early disease detection.

### 4.6. Limitations

The limitations include the automatic machine learning system used to differentiate AD patients from healthy subjects, which is still developing, and the fact that our participants consisted of only Taiwanese people; therefore, an extensive application still needs to be further evaluated. Our study includes a small participant number and is a cross-sectional study, thus a larger, longitudinal study may provide more information on the issue of AD patients’ facial characteristics.

OpenFace uses Conditional Local Neural Fields (CLNFs) for facial landmark detection and tracking. A CLNF’s performance depends on dataset diversity. Diverse datasets enhance adaptability, while small or biased datasets may hinder performance. Fine-tuning, data augmentation, and context-specific evaluation are often needed. In our study, we controlled image quality, facial pose, and expression. However, our dataset was limited in diversity and scale.

Additionally, overstating the connection between facial appearance and diseases can lead to biases. The current diagnostic criteria continue to be the basis for AD diagnosis. The early and accurate detection of AD remains an unmet need, and we propose that facial asymmetry could potentially serve as a supplementary tool in the future, alongside emerging early diagnostic methods for AD.

## 5. Conclusions

AD patients exhibited increased facial asymmetry compared to age- and gender-matched individuals without dementia. Our analysis encompassed a total of 29 facial landmarks, revealing that 20 pairs of LSADs in AD patients were significantly larger than those in the control group. Asymmetry in facial features, including face edges, eyebrows, eyes, nostrils, and mouth, contributed to distinguishing AD from non-dementia controls. Specifically, differences in face edge pairs 1 through 8, as well as eyebrows pairs 9, 10, and 13, nostrils pairs 14 and 15, eye pairs 16 through 19, and mouth pairs 22, 24, and 26, were statistically significant, with *p*-values ranging from 0.001 to 0.041. Furthermore, the total LSAD was notably higher in AD patients compared to controls, with a *p*-value of 0.003. Facial asymmetry may potentially serve as a tool for early AD detection in the future. However, it is essential to validate these findings in larger and more diverse cohorts.

## Figures and Tables

**Figure 1 biomedicines-11-02802-f001:**
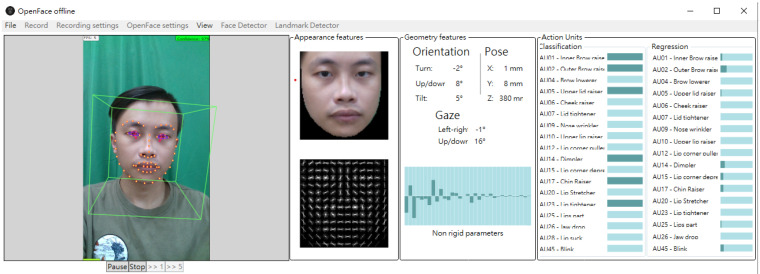
The image registration process example of OpenFace.

**Figure 2 biomedicines-11-02802-f002:**
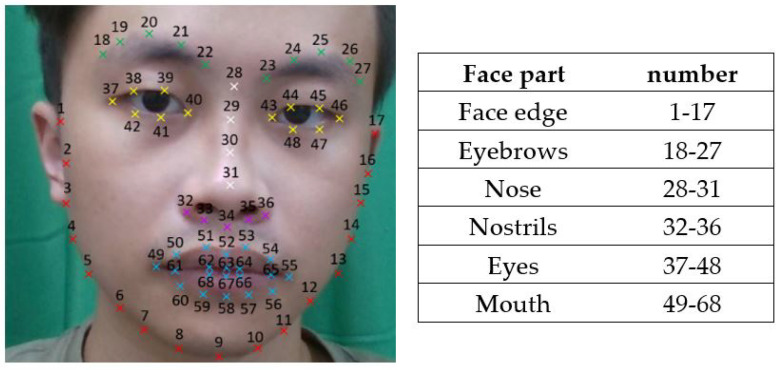
Facial feature points distribution map by OpenFace.

**Figure 3 biomedicines-11-02802-f003:**
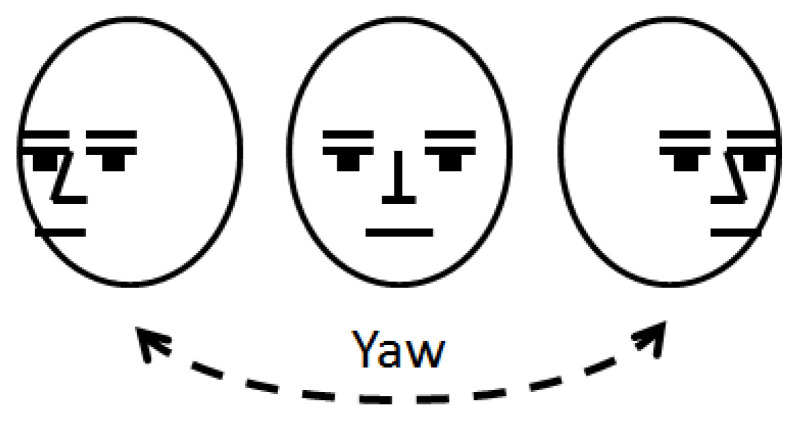
Yaw angle.

**Figure 4 biomedicines-11-02802-f004:**
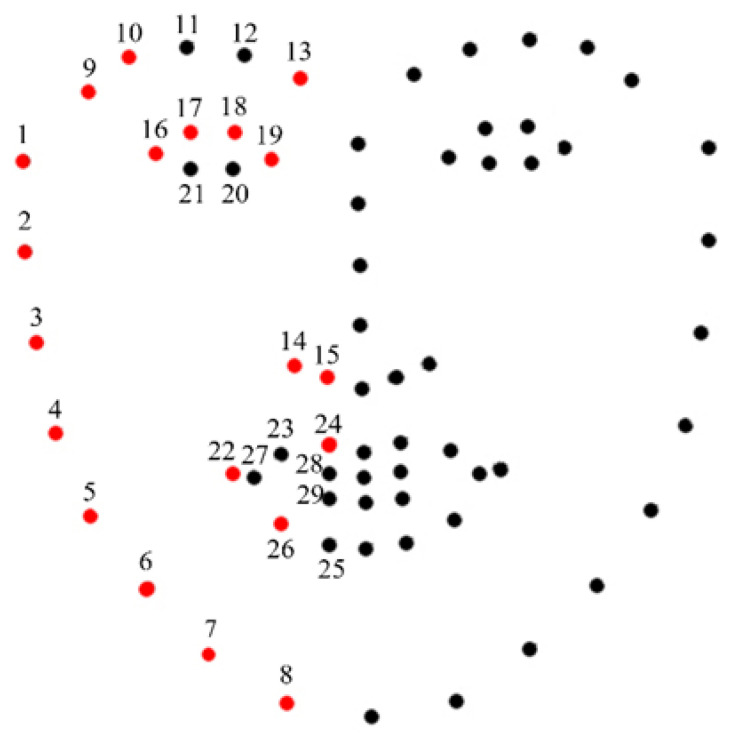
Facial feature point pairs of LSADs with a significant difference between AD patients and controls are marked as red.

**Table 1 biomedicines-11-02802-t001:** Demographic characteristics of recruited participants.

	AD	Controls	*p*-Value
Age, years (mean ± SD)	72.2 ± 4.9	71.8 ± 7.4	0.15
Gender (patient No., (%))			0.19
Male	57 (38%)	46 (31%)	
Female	93 (62%)	104 (69%)	
CDR (%)			
0	2 (1.4%)		
0.5	49 (34.8%)		
1.0	76 (53.9%)		
2.0	14 (9.9%)		
3.0	0		
CDR-SB (mean ± SD)	4.8 ± 2.9	1.5 ± 1.4	
MMSE (mean ± SD)	21.0 ± 4.4	24.5 ± 3.7	
CASI (mean ± SD)	65.7 ± 14.2	83.7 ± 10.5	

CDR: Clinical Dementia Rating. MMSE: Mini-Mental State Examination. SD: standard deviation.

**Table 2 biomedicines-11-02802-t002:** The facial feature point pairs of LSADs, calculated with *t*-test between AD and controls.

Face Part	Pair	AD	Controls	*p*-Value
Face edge	1	0.71 ± 0.51	0.63 ± 0.53	0.003 *
	2	0.43 ± 0.30	0.40 ± 0.37	0.006 *
	3	0.47 ± 0.34	0.39 ± 0.33	0.009 *
	4	0.56 ± 0.37	0.46 ± 0.37	0.012 *
	5	0.57 ± 0.39	0.48 ± 0.37	0.010 *
	6	0.51 ± 0.33	0.44 ± 0.32	0.003 *
	7	0.41 ± 0.31	0.37 ± 0.31	0.041 *
	8	0.34 ± 0.35	0.25 ± 0.24	0.007 *
Eyebrows	9	0.25 ± 0.19	0.22 ± 0.15	0.005 *
	10	0.17 ± 0.12	0.16 ± 0.12	0.004 *
	11	0.24 ± 0.18	0.22 ± 0.18	0.059
	12	0.30 ± 0.22	0.27 ± 0.20	0.053
	13	0.29 ± 0.20	0.25 ± 0.18	0.001 *
Nostrils	14	0.37 ± 0.23	0.31 ± 0.22	0.013 *
	15	0.44 ± 0.25	0.39 ± 0.26	0.016 *
Eyes	16	0.12 ± 0.10	0.09 ± 0.06	0.024 *
	17	0.15 ± 0.12	0.11 ± 0.09	0.001 *
	18	0.13 ± 0.10	0.11 ± 0.09	0.0004 *
	19	0.12 ± 0.07	0.10 ± 0.07	0.0009 *
	20	0.32 ± 0.11	0.32 ± 0.09	0.209
	21	0.29 ± 0.11	0.28 ± 0.08	0.747
Mouth	22	0.13 ± 0.09	0.11 ± 0.08	0.023 *
	23	0.31 ± 0.17	0.27 ± 0.17	0.054
	24	0.43 ± 0.23	0.37 ± 0.23	0.035 *
	25	0.33 ± 0.19	0.27 ± 0.17	0.083
	26	0.25 ± 0.16	0.21 ± 0.15	0.042 *
	27	0.82 ± 0.27	0.79 ± 0.25	0.936
	28	0.52 ± 0.15	0.55 ± 0.13	0.645
	29	0.36 ± 0.20	0.31 ± 0.19	0.069
Sum of LSAD		10.33 ± 4.39	9.14 ± 4.45	0.003 *

* *p* < 0.05.

## Data Availability

Third party data: Restrictions apply to the availability of these data. The data were obtained from Department of Mechanical and Electromechanical Engineering, National Sun Yat-Sen University, Kaohsiung, Taiwan and are available from Chen-Wen Yen and Yuan-Han Yang with the permission of Department of Mechanical and Electromechanical Engineering, National Sun Yat-Sen University, Kaohsiung, Taiwan.

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
