# Peer review of "Analyzing Facial Asymmetry in Alzheimer’s Dementia Using Image-Based Technology"

_biomedicines, 2023, doi:10.3390/biomedicines11102802_

Round 1

Reviewer 1 Report

(1)Based on the content of the paper and the theme of the journal, it appears that they do not align closely. Therefore, I would suggest considering alternative journals.

(2) The structure of the abstract and the body of the article exhibit some differences. According to the abstract, accelerated brain aging is an important assessment method for AD. As facial images are relatively easier to obtain compared to brain images, facial asymmetry serves as an effective alternative approach. However, the main body of the article lacks necessary background description on the relationship between accelerated brain aging and AD. It is recommended to include a background paragraph providing context. The following are some relevant papers:

Brainage as an important biomarker:

Cole JH, Franke K. Predicting Age Using Neuroimaging: Innovative Brain Ageing Biomarkers. Trends Neurosci. 2017 Dec;40(12):681-690.

Franke K, Gaser C. Ten Years of BrainAGE as a Neuroimaging Biomarker of Brain Aging: What Insights Have We Gained? Front Neurol. 2019

The relationship between accelerated brain aging and AD

Huang W, Li X, Li H, Wang W, Chen K, Xu K, Zhang J, Chen Y, Wei D, Shu N, Zhang Z. Accelerated Brain Aging in Amnestic Mild Cognitive Impairment: Relationships with Individual Cognitive Decline, Risk Factors for Alzheimer Disease, and Clinical Progression. Radiol Artif Intell. 2021 Jun 23;3(5):e200171.

Franke K, Ziegler G, Klöppel S, Gaser C; Alzheimer's Disease Neuroimaging Initiative. Estimating the age of healthy subjects from T1-weighted MRI scans using kernel methods: exploring the influence of various parameters. Neuroimage. 2010 Apr 15;50(3):883-92.

An explanatory framework for the phenomenon of heterogeneity  in brain aging:

Lin, L.; Xiong, M.; Jin, Y.; Kang, W.; Wu, S.; Sun, S.; Fu, Z. Quantifying Brain and Cognitive Maintenance as Key Indicators for Sustainable Cognitive Aging: Insights from the UK Biobank. Sustainability 2023, 15, 9620.

(3)The existence of significant differences in facial features among different ethnic groups raises questions about the generalizability of research findings. When examining the generalizability of research findings related to facial features, it is crucial to consider the limitations and potential biases inherent in the studies. Many facial feature studies have primarily focused on populations of specific ethnic backgrounds, often those of European descent. As a result, the applicability of these findings to other ethnic groups, including Taiwanese individuals, may be uncertain. It is advisable to incorporate research about the differences between Taiwanese individuals and other ethnic groupsin the Discussion section.

(4) In order to gain a comprehensive understanding of this study, it is necessary to include the diagnostic criteria for AD.

(5) In section 3.2, it is recommended to present the information using a table format for enhanced clarity and organization.

(6)It is recommended to conduct a correlation analysis between significant landmarks  and cognitive measures. This helps to determine whether facial characteristics can serve as indicators of disease development.

Reviewer 2 Report

The paper by Chien and colleagues entitled ‘Analysis of Facial Asymmetry in Alzheimer's Dementia Using Artificial Intelligence-Assisted Image-Based Methods’ aimed to investigate facial asymmetry as a potential indicator for Alzheimer's disease (AD) and differentiate AD patients from non-dementia controls. The study involved 300 Taiwanese participants, including 150 AD patients and 150 controls without dementia. Three-dimensional photogrammetry and landmark detection were used to measure facial asymmetry. The results showed that AD patients had significantly greater facial asymmetry compared to controls in various facial landmarks, including face edges, eyebrows, eyes, nostrils, and mouth. This suggests that facial asymmetry could be a potential tool for early detection of AD. However, further research with larger sample sizes and longitudinal studies is needed to validate these findings.

Overall, I find the objective presented in this article to be quite intriguing, and the authors' insightful observations on this relevant subject matter could capture the attention of Biomedicines readers. However, there are certain points worth addressing, including specific comments and essential evidence required to bolster the author's argument. These adjustments are necessary to enhance the manuscript's quality, suitability, and overall readability before it can be published in its current state. In conclusion, I recommend publication of this research article once the author has thoughtfully incorporated the feedback and suggestions outlined below.

- According to the Journal’s guidelines, your research manuscript should comprise the Front matter (title, authors list, affiliations, abstract and keywords), Introduction, Materials and Methods, Results, Discussion and Conclusions section, and the Back matter (Supplementary Materials, Acknowledgments, Author Contributions, Conflicts of Interest, References). 

- The title is somewhat long and includes technical terms like "Artificial Intelligence-Assisted Image-Based Method." While these terms accurately describe the methodology used, they may make the title less accessible to a general audience. Consider simplifying or clarifying the title without sacrificing accuracy. Also, it mentions both "Alzheimer's Dementia" and "Alzheimer's Disease." While these terms are often used interchangeably, using just one term consistently throughout the title may improve readability.

- I strongly recommend to include a graphical abstract that summarizes the main findings of the manuscript.

- Abstract: According to the Journal’s guidelines, the abstract should be a total of about 200 words maximum. The abstract should be a single paragraph and should follow the style of structured abstracts, but without headings. Correct the actual one.

- The introduction appears to lack clear organization and structure. It jumps between various topics related to Alzheimer's disease, facial asymmetry, and methods without a clear flow. Consider reorganizing the information to provide a more logical and structured progression of ideas. Also, it would be beneficial to include a brief mention of the neural substrates and brain regions known to be affected by Alzheimer's disease (AD) to establish a stronger connection between AD and the study's focus on facial asymmetry. This addition would help readers understand the neurological basis for investigating facial asymmetry in AD patients and the potential implications for early detection and management of the disease. Please consider incorporating a concise paragraph or sentence that highlights the neural substrates impacted by AD within the introduction. This would not only enhance the clarity of the research context but also emphasize the importance of studying facial morphology as a potential marker of AD-related changes in brain structure and function [1-3].

- Regarding the clinical data collection, consider explaining why the specific camera (Intel RealSense D435) was chosen for capturing 3D facial images. Was it because of its accuracy, availability, or other factors? I also suggest to clarify if there were any specific criteria for selecting the 30 images captured for each participant. Were these selected randomly, or were there specific guidelines for image selection?

- Face Landmark Detection and Pre-processing: Provide some context or reasoning behind the choice of Openface for detecting facial landmarks. Why was this particular toolkit chosen over others? When discussing the rules for replacing landmark depth measurements, provide a bit more detail or examples for better comprehension.

- Normalization and Procrustes Analysis: Consider providing a brief explanation of what the Procrustes method entails and how it helps align facial images. This would be especially useful for readers who may not be familiar with this technique. When discussing the determination of the depth coefficient 'c,' it would be beneficial to include a summary of the results obtained from testing different values of 'c.' This could help readers understand why 'c' was ultimately chosen as 0.016.

- Finally, the discussion could benefit from better organization and subheadings to clearly separate different aspects of the study. For example, authors could have subsections for the study's objectives, key findings, method comparisons, strengths, and limitations, as well as provide a more comprehensive review of relevant literature. Discuss how your findings align or differ from existing research and theories. Finally, in my opinion, this section should delve deeper into the interpretation of the results. What might be the underlying reasons for the observed facial asymmetry in AD patients? Are there potential clinical implications? Consider discussing the possible biological or neurological mechanisms.

- References: Authors should consider revising the bibliography, as there are several incorrect citations. Indeed, according to the Journal’s guidelines, they should provide the abbreviated journal name in italics, the year of publication in bold, the volume number in italics for all the references. Also, some of the references are out of date:  please cite references from the last 10 years, particularly references from the recent 5 years.

I hope that, after careful revisions, the manuscript can meet the journal’s high standards for publication. I declare no conflict of interest regarding this manuscript.

Best regards,

Reviewer

References: 

1. https://doi.org/10.3389/fnmol.2023.1217090

2. https://doi.org/10.3390/biomedicines11051248

3. DOI: 10.3390/biomedicines11030945

Minor editing of English language required.

Reviewer 3 Report

The paper aimed to determine if facial asymmetry could serve as an indicator of Alzheimer’s dementia (AD). By employing a three-dimensional camera and machine learning through OpenFace to detect facial landmarks, the research compared facial images of 150 Alzheimer’s patients to 150 age- and gender-matched controls. The results revealed a statistically significant increase in facial asymmetry among the AD group, particularly around the face edge, eyebrow, eyes, nostrils, and mouth. Despite these findings, the study's limitations included its focus on Taiwanese participants and its relatively small sample size, with the authors suggesting that further expansive studies would provide deeper insights. Upon thorough examination of the submitted manuscript, several pressing concerns arose, warranting critical scrutiny.

·        The introduction posits a vague and speculative question regarding the connection between AD and facial morphology. For a study of this kind, a clear and concrete hypothesis would have been more suitable.

·        The method section describes the use of OpenFace for landmark detection, but it omits elaboration on the image registration technique used. Furthermore, the decision criteria for selecting 29 pairs out of 68 facial landmarks for LSAD are unclear.

·        The manuscript does not clarify whether facial asymmetry could be a cause or an effect of Alzheimer’s dementia. It's imperative to avoid suggesting causation when the study is only capable of pointing to a correlation. The authors need to discuss the problem of facial symmetry evaluation not only for AD, but also for other conditions characterized by face asymmetry such as facial palsy (Few-shot learning with a novel voronoi tessellation-based image augmentation method for facial palsy detection), as well as for normal (healthy) people (Assessing Facial Symmetry and Attractiveness using Augmented Reality).

·        The exclusive selection of Taiwanese participants dramatically narrows the study’s applicability. The paper fails to address potential racial and ethnic differences in facial asymmetry, which is a glaring oversight when claiming a connection between AD and facial asymmetry.

·        The dataset is imbalanced both in terms of gender and condition, which may introduce bias. Could it be remedied using for example, image augmentation or GAN networks to generate synthetic faces and balance the dataset? You may consider Improvement of Facial Beauty Prediction Using Artificial Human Faces Generated by Generative Adversarial Network.

·        The paper claims their study's strengths include utilizing a state-of-art machine learning algorithm and a systematic approach with a 3D camera, yet it offers no comparative data or references to justify these as “strengths”.

·        The limited participant number and the cross-sectional nature of the study significantly reduce its power and generalizability. For a claim as potentially groundbreaking as early detection of AD through facial asymmetry, a larger and more diverse sample set is crucial.

·        The article redundantly iterates that facial asymmetry may aid in early dementia detection. One thorough statement would have sufficed.

·        Even if facial asymmetry is linked to AD, the paper should discuss the feasibility and practicality of this method for real-world implementation. There are various challenges and ethical considerations in using facial asymmetry as a potential diagnostic tool that the paper seems to overlook.

·        The paper mentions a machine learning model but does not discuss its potential biases. The real-world utility of such a model hinges on its generalizability across diverse datasets, which the paper fails to address.

·        While it's commendable to embark on uncharted territory, there's an overemphasis on the paper being "the first study to differentiate AD patients from non-dementia people through objective evaluation of facial asymmetry." Novelty alone does not ensure validity or value.

The paper requires substantial revision and fortification of its claims. Until these concerns are adequately addressed, its scientific value remains contentious.

Round 2

Reviewer 1 Report

The areas of uncertainty in the manuscript have been largely addressed, and I recommend its publication.

Author Response

Thank you for your review and feedback. 

Reviewer 2 Report

Dear Authors,

I am pleased to acknowledge that you have indeed addressed all of my concerns and queries in a clear and precise manner. Your responses have provided valuable insights into the modifications made to the manuscript in light of my comments. It is evident that you have taken great care to ensure that the revised manuscript aligns more closely with the scientific rigor expected for publication in Brain Sciences.

Upon reviewing the updated version, I find that the inclusion of the additional studies has indeed enriched the understanding of neural substrates and brain regions known to be affected by Alzheimer's disease (AD) to establish a stronger connection between AD and the study's focus on facial asymmetry. The provided studies contribute significantly to the comprehensiveness of the section. However, in order to provide a more holistic view of the neural structures, I believe there's still an opportunity to expand upon certain factors. Specifically, the discussion of the neural mechanisms underlying the phenomena being investigated could offer a deeper insight into the mechanisms at play (https://doi.org/10.1111/acps.13602; https://doi.org/10.3390/biomedicines11051248; DOI: 10.3390/biomedicines11030945). 

I want to reiterate my appreciation for your responsiveness and willingness to consider these suggestions. I believe that this minor revision will significantly enhance the quality and impact of the Introduction section. 

Thank you once again for your dedication to improving the manuscript. I look forward to seeing the continued progress.

Best regards,

Reviewer

Author Response

Reviewer 2

Specifically, the discussion of the neural mechanisms underlying the phenomena being investigated could offer a deeper insight into the mechanisms at play (https://doi.org/10.1111/acps.13602; https://doi.org/10.3390/biomedicines11051248; DOI: 10.3390/biomedicines11030945). For introduction.

Ans: Thanks. We have made the necessary revisions.

Reviewer 3 Report

The conclusions section should be extended. The claims should be supported with the numerical findings from the study (such as p-values).

Author Response

Reviewer 3

The conclusions section should be extended. The claims should be supported with the numerical findings from the study (such as p-values).

Ans: Thanks. We have made the necessary revisions.